artificial intelligence/human-computer interaction

data clustering, human-computing, crowdsourcing, games

**Author for correspondence:**
Jérôme Waldispühl
e-mail: jerome.waldispuhl@mcgill.ca

# Human-supervised clustering of multidimensional data using crowdsourcing

Alexander Butyaev[1], Chrisostomos Drogaris[1], Olivier Tremblay-Savard[2] and Jérôme Waldispühl[1]

[1]School of Computer Science, McGill University, Montréal, Canada
[2]Department of Computer Science, University of Manitoba, Winnipeg, Canada

 JW, 0000-0002-2561-7117

Clustering is a central task in many data analysis applications. However, there is no universally accepted metric to decide the occurrence of clusters. Ultimately, we have to resort to a consensus between experts. The problem is amplified with high-dimensional datasets where classical distances become uninformative and the ability of humans to fully apprehend the distribution of the data is challenged. In this paper, we design a mobile human-computing game as a tool to query human perception for the multidimensional data clustering problem. We propose two clustering algorithms that partially or entirely rely on aggregated human answers and report the results of two experiments conducted on synthetic and real-world datasets. We show that our methods perform on par or better than the most popular automated clustering algorithms. Our results suggest that hybrid systems leveraging annotations of partial datasets collected through crowdsourcing platforms can be an efficient strategy to capture the collective wisdom for solving abstract computational problems.

## 1. Introduction

Data clustering is a central task in many computer analysis techniques, routinely used in data mining and unsupervised machine learning [1,2]. Techniques are constantly being improved [3,4], but they all face a common bottleneck. With diverse shapes, densities or sizes, the definition of a cluster is intrinsically ambiguous, subject to different interpretations, and often requires domain knowledge for producing accurate annotations (e.g. gating for flow cytometry data analysis [5]). Even though progress is still being made towards automated feature selection [6] and modelling of noisy data [7], there is no single universally accepted mathematical measure that can unambiguously determine the

existence or boundaries of a cluster. In these circumstances, the performance of algorithms varies across datasets and may still require validations from experts.

In fact, the clustering task is rooted in the field of psychology [8], where the decision to group data or objects together answers to principles of grouping also referred to as the Gestalt laws of grouping [9]. The development of algorithms capable of capturing this information is a long-standing problem [10], and there is to our knowledge no framework allowing us to semi-automatically harness the collective perception of crowds of humans. The goal of this contribution is thus to offer a proof of concept that human-computing and crowdsourcing techniques can be applied to address the challenge of clustering abstract data with various sizes, shapes and densities.

Among the various measures previously introduced to estimate the quality of a cluster annotation [11], Silhouette [12], Dunn [13], SDbw [14], and Modularity [15] emerge as the most popular. This situation resulted in the development of a broad variety of algorithms and metrics [16], but as mentioned above, there is still a lack of a clear consensus. In fact, the presence (or absence) of a cluster ultimately results from an agreement between multiple individuals and preferentially data analysis experts with domain knowledge [17,18].

Unfortunately, manual expert annotation is not scalable to large datasets. But crowdsourcing can be an attractive alternate solution to perform a large number of repetitive tasks, with the underlying assumption that the aggregation of a sufficiently large number of answers from non-experts could approximate expert behaviours [19–21]. Such an approach opens the door to large-scale experiments allowing us to calibrate human perception of clusters against the performance and behaviour of some of the most popular clustering algorithms. However, it is never obvious if using crowdsourced data will result in an improvement of annotations or rather add some noise that will deteriorate the quality of the predictions.

We thus aim to cluster abstract data (i.e. points with coordinates) using crowdsourcing. But the dimensionality of the dataset may radically change the nature of the problem. The visual perception of a cluster is an intuitive concept in low-dimensional spaces (i.e. up to three dimensions), but much less so in higher dimensions when a single user cannot apprehend the full distribution of the data. In addition to the unavoidable challenge of *the curse of dimensionality* [22], it is unclear if a crowdsourcing approach will scale well with higher-dimensional datasets or could even compete with state-of-the-art automated methods.

In this paper, we present Colony B (http://colonyb.com), a mobile human-computing game for collecting human input for visually clustering multidimensional data. We introduce two clustering algorithms, hubCLIQUE and CloCworks, using aggregated crowd solutions to either (i) improve the accuracy of seminal methods, or (ii) build clusters from scratch. We conducted two short-term experiments using synthetic and real-world datasets. Using accumulated answers of the Colony B players, we benchmark both algorithms against popular automated clustering methods, and show that hubCLIQUE and CloCworks performed on par with or better than other conventional approaches. Overall, this work is a proof of concept that human-computing and crowdsourcing are promising technologies for the development of semi-automated clustering methods of abstract data.

## 2. Related work

Human-computing and crowdsourcing have been used in many different ways recently to help with clustering items such as words, documents, and images. The different approaches that are used can be divided into three main categories: crowd clustering, interactive clustering, and assisted clustering.

Crowd-clustering approaches typically employ the crowd throughout the clustering process. Cascade [23] is such an approach that focuses on not only clustering items but also creating a full taxonomy. It accomplishes its goal by using three types of tasks to be crowdsourced: asking the workers to (i) generate categories for a certain number of items, (ii) select the best category (out of many) for a specific item, and (iii) choose all categories that fit one specific item. Deluge [24] is a refinement of Cascade that uses decision theory and machine learning to reduce the amount of human work required by specifically finding microtasks that can maximize information gain. The crowd synthesis work [25] showed the advantages of using a more global approach by showing more information to the workers and using iterative clustering, which provides workers with an overview of the previously created categories.

Interactive clustering generally refers to the combination of users and machine-learning algorithms to accomplish the clustering task. In this type of human-in-the-loop clustering, the users indicate to the system some kind of constraints that allow the machine to learn metrics. The information provided by

the users can be pairwise labels of the must-link/cannot-link type [26,27], which indicate to the system what elements must be or should not be in the same resulting clusters. Other systems based on split/merge constraints allow the users to specify which pre-computed clusters should be merged together or split by the next iteration of the clustering algorithm [28]. Another interactive clustering model named Grouper proposed a refine-and-lock model instead, in which the human is involved throughout the clustering process [29]. In Grouper, an initial cluster computed algorithmically is presented to the user, who can then reassign items to other clusters and lock clusters that are not to be modified again. Then the machine uses the information from the users to recalibrate its distances and reclusters the still unlocked items. These steps are repeated until all the clusters are locked by the user. The Alloy system [30] presents a *cast and gather* approach, in which users provide 'casts' of human judgement (by the means of three different types of tasks) which are then 'gathered' by the machine-learning algorithm. In other words, the general idea is to present clips of texts to users, who will then identify the clips that are not representing the same information and highlight keywords in these clips. Then the system presents similar clips based on the identified keywords and the users have to label the search results as similar or different to the initial clip. This information is then used to train a machine-learning algorithm that will deal with the majority of the data.

The idea of assisted clustering came from an observation that it seemed to be more intuitive for the users to build clusters and specify must-belong and cannot-belong constraints between items, rather than the traditional must-link/cannot-link relationships [31]. From the perspective of assisted clustering, the role of the machine is to learn to classify and suggest items that might be relevant for the user to add to a certain cluster.

As far as we know, crowdsourcing has rarely been used to help identify clusters on a two-dimensional visual representation of data points. The closest example we could find in the literature was the work on interactive *t*-SNE clustering of Bond *et al.* [32]. *t*-SNE, a visualization approach for high-dimensional data that creates an embedding of the data points on a two-dimensional (or three-dimensional) space, focuses on putting very similar data points close together in the final embedding [33]. *t*-SNE can however struggle with the clustering of boundary points, and Bond *et al.* [32] created an interface that allows users to view the *t*-SNE clustering process in real time and move data points. The new positions of these data points are then sent to the algorithm, which incorporates these changes and continues its work.

# 3. System overview

## 3.1. Motivation

The goal of this paper is to explore the potential of crowdsourcing approaches for clustering complex datasets with a casual game for non-experts. A major challenge with automated clustering methods is the absence of a gold-standard metric to assess the existence of a cluster. Because the validation of a cluster ultimately results from an agreement of several experts, here we aim to explore the potential of semi-automated strategies using annotations on partial data from non-experts to guide the clustering process. Instead of showing the crowd raw data, we choose to project data points on a two-dimensional game board and make use of the human brain's intuitive ability to spot clusters.

We emphasize the relevance of this approach for high-dimensional datasets that are affected by the curse of dimensionality and for which the robustness of classical algorithms is challenged [34]. Here, we aim to use crowdsourced annotations to improve confidence in the quality of the predictions regardless of the dimension in which the clusters are residing.

While there are tools for recruiting paid crowd workers to complete micro tasks [35,36], we chose to use a game because of the versatility and complexity of tasks we can present but also the scalability of the approach. Indeed, if the most popular crowdsourcing platforms gather thousands of workers [37], the mobile consumer market is constantly growing [38] with more than 2.8 billion monthly active mobile users worldwide spending over 3 h per day on their device [39]. Moreover, curiosity and games have been shown to be similarly efficient as conventional monetary-based platforms in several different applications [40,41].

## 3.2. Description of the game

We designed an online human-computing game for mobile devices that is freely and publicly available on all major platforms. Each *puzzle* of Colony B consists of multiple consecutive two-dimensional screens in which dots (data points) are presented to the user, who has to select the dots that form the best cluster that they perceive. Each individual two-dimensional screen of a certain puzzle (which we call a *stage*) is a

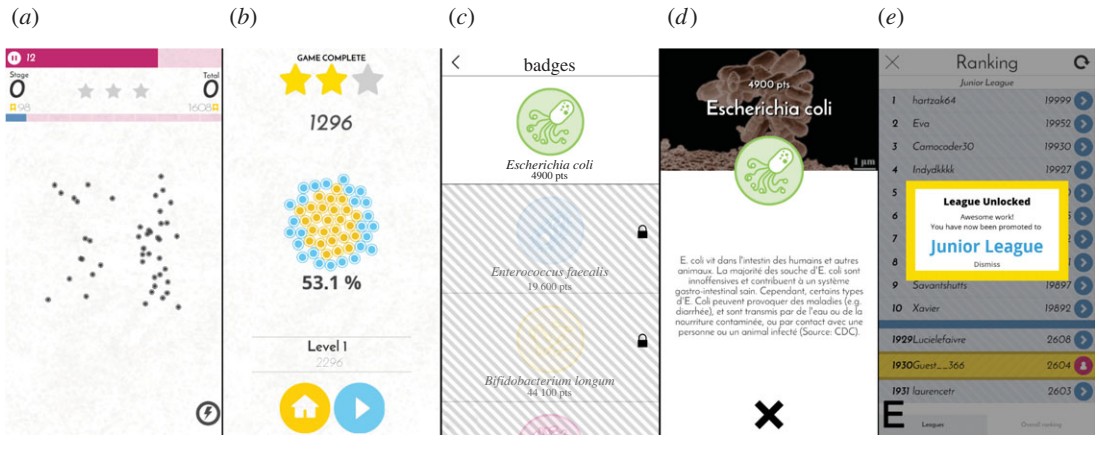

**Figure 1.** Illustration of multiple scenes of Colony B. (*a*) Clustering panel with stage/total scores and puzzle progress, (*b*) end-game screen showing the progress towards a new badge discovery, (*c*) list of available thematic badges, (*d*) educational information related to a badge, and (*e*) leaderboard with multiple leagues.

visualization of the same data points from a different angle. In other words, since the dataset has multiple dimensions, each puzzle is divided into stages that represent pairs of dimensions at a time. This strategy allows us to analyse multidimensional datasets while presenting the data to players in a convenient and simple two-dimensional interface.

Before going further, it is worth noting that the game is framed as a citizen science game in which the players cluster bacteria. While this aspect is not relevant for this contribution, it aims to prepare the application of our technology to the analysis of real biological datasets and educate participants about microbiology through the game.

On each stage (figure 1*a*), players have to select a single group of dots and then automatically move on to the following stage. The solution (on each stage and for the entire puzzle) is evaluated using the scoring function presented below (see §3.4), and the resulting game points are added to the player's profile (figure 1*b*).

To reward the players, thematic badges (which include some educational content that is related to the data that is being analysed; see figure 1*c*,*d*) are given to the players once they reach certain thresholds of accumulated game points. Players are also divided into different leagues according to their experience level with the game, and game points also allow players to get promoted to higher leagues.

To speed up the game, a timer is included on each stage (see the time slider at the top of the screen in figure 1*a*), which plays a role with bonus points that players receive according to how much time is left.

To ensure players understand the goal of the game, we designed a step-by-step interactive tutorial that explains each aspect of the game with immediate feedback on every action of the player.

An example of the process of solving one puzzle (composed of many stages) is partially shown in figure 2. Figure 2*a*–*c* illustrates the first stage of the puzzle: (*a*) the player is first presented with the data points of the stage, (*b*) selects a cluster on the screen, (*c*) receives feedback from the system (score), and then moves on to the next stage. Figure 2*d*,*e* shows the process for the following stage. Since the dots on consecutive stages are the same data points viewed from different angles (i.e. different pairs of dimensions), the system colours the dots that were selected by the player on the previous stage in blue. Bonus game points are gained for selecting again the same data points in consecutive stages (see §3.4 for more details). This is how the game links together the multiple dimensions that are being viewed on separate stages. The general idea here is to give extra value to clusters of data points that are present in as many different pairs of dimensions as possible.

## 3.3. Adapting the raw multidimensional data

Consider a raw multidimensional dataset with $N$ points in a $F$-dimensional space (i.e. composed of $F$ features).

As mentioned earlier, our game interface is limited to two dimensions (orthogonal projections) at a time, and different pairs of dimensions are presented in consecutive stages. However, it is still not feasible to ask users to process all $\binom{F}{2}$ distinct two-dimensional projections of the dataset. After preliminary analyses with different numbers of the stages per puzzle, we found the optimal number

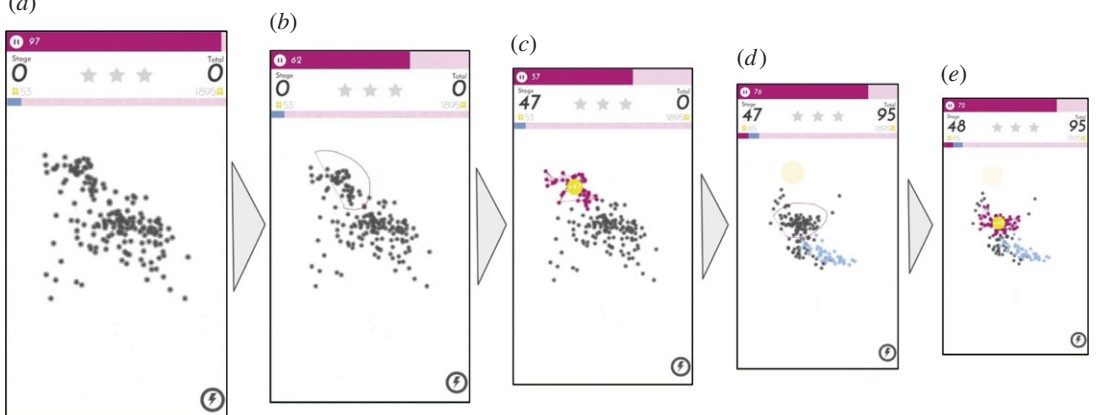

**Figure 2.** Game flow showing the first two stages and the process of puzzle solving. The player: (*a*) is presented with initial stage data; (*b*) makes a selection of the most representative group of dots on the mobile screen; (*c*) receives feedback from the game (score); and (*d*) is presented with the next stage data. It also shows in blue the points selected by the player on the previous stage. The player selects a group of dots for this stage and (*e*) receives feedback from the game.

of stages to be 15. Therefore, we had to limit the dimensionality of the datasets to six, to make sure that all possible pairs of dimensions can be used in a puzzle.

To accommodate for this restriction, many dimensionality reduction techniques, feature selection techniques, or iterative random sampling of features exist to address multidimensional datasets [42]. In most cases, the choice of the strategy must consider the nature of the data, the distance metrics used, and other data-related properties (see §5 for a detailed description of how datasets were pre-processed for this study).

Another aspect to keep in mind, since the game is targeting mobile platforms, is that we are limited by the computational resources and rendering capabilities of a mobile device. Concretely, it means that it is rarely possible to show all the data points of an entire dataset in a puzzle. For this reason, we pre-generated a set of puzzles, each containing a fixed number of randomly selected data points. We defined this number of points as the maximum possible number that allows a user to interact smoothly with our mobile application using an average mobile device (we used an iPhone 4S). As a result, a single puzzle often is a very sparse representation of the dataset, but by adjusting the number of puzzles we can control its overall coverage.

## 3.4. Scoring player solutions

Each solution collected from a player corresponds to a human clustering result for every stage of a puzzle. As such, we designed a multiphase scoring strategy that takes into account the individual stage solutions as well as information extracted from transitions between stages. It is a generally forgiving scoring method that allows enough variety in the player solutions, and all the while it penalizes random answers.

First, each stage is evaluated using three clustering validation indices: (i) Silhouette, (ii) SDbw, and (iii) Dunn [16]. Since our scoring function requires a comparison of the indices, and since they each provide different ranges of values, we propose a heuristic to normalize them. We first recorded sample distributions of values for each index on about 30 different clusters (found by KMeans and spectral clustering) on a total of 15 000 randomly selected stages of each of the studied datasets (described in §5). This step is done once, so the produced distributions are then available for all datasets uploaded to the Colony B system. Then, we evaluated each human solution by calculating the resulting score of the three indices and comparing them with the reference distributions of scores. Basically, for each index, the proportion of the sample clusters in the distribution that showed a worse score than the current human solution is calculated. This procedure equally scales all three index measures in the interval [0, 1].

The best value obtained after the normalization step (i.e. the best proportion out of the three indices) is assigned to what we call the quality score $Qual(\mathcal{S}_i)$ for this specific stage $\mathcal{S}_i$. As mentioned earlier, to link together the different pairs of dimensions on consecutive stages, we introduce a *conservation* score

$Cons(\mathcal{S}_{i-1}, \mathcal{S}_i)$, i.e. the number of points in the proposed cluster that were also selected in the previous cluster. We also use a density correction coefficient $\mathcal{DCC}$ to shift players' attention towards significantly dense areas. It also aims at reducing potentially spontaneous or random behaviour of players trying to quickly go through the stages. The final scoring function for a player's stage evaluation is

$$\text{Score}(\mathcal{S}_i) = \big(\alpha \times \text{Qual}(\mathcal{S}_i) + (1 - \alpha) \times \text{Cons}(\mathcal{S}_{i-1}, \mathcal{S}_i)\big) \times \frac{1 + \mathcal{DCC}}{2},$$

where $\alpha$ is a variable in the range of [0, 1] that controls how much weight is given to the quality score with respect to the conservation score (we used 0.8 in this study), and $\mathcal{DCC}$ is a shifted density value obtained using kernel density estimation [43]. The magnitude of a density shift defines the sensitivity of the score to multiple dense areas within a single selection.

The final score of the entire puzzle is then summed across all the individual stages of that puzzle.

# 4. Methods

While most clustering algorithms produce robust results in low-dimensional spaces, only a few perform adequately in multidimensional spaces where the curse of dimensionality becomes noticeable [34]. In practice, subspace clustering has often been found to yield the best results by identifying clusters that are hidden in specific subspace(s) while presented with noise from other dimensions. Our methods aim to address this challenge and use human input to augment the performance of classical algorithms.

In this section, we describe two novel semi-automated clustering algorithms for multidimensional data leveraging the information embedded in annotations collected from Colony B participants. The first one, **hu**man-**b**ased **CL**ustering **In QUE**ue (hubCLIQUE), uses a bottom-up approach that generalizes the seminal CLIQUE algorithm [44]. The second one, **C**lustering **O**f **C**rowdsourced net**works** (CloCworks), applies a community detection strategy to identify groups of answers in agreement to generate the clusters.

## 4.1. hubCLIQUE

CLIQUE (**CL**ustering **In QUE**ue) [44] is one of the first, and still one of the most popular, algorithms developed for clustering high-dimensional datasets. It uses grid-based and density-based approaches to identify dense areas in lower-dimensional spaces and progressively expands the candidate clusters in higher dimensions. This strategy is flexible enough to easily incorporate additional information extracted from a human input. We call this algorithm hubCLIQUE, a bottom-up subspace clustering approach guided by crowdsourced solutions collected by Colony B.

### 4.1.1. Data initialization and metrics

CLIQUE starts with computing seeds for the clusters. This initialization is done by finding dense regions in low-dimensional spaces (i.e. the primitive subspaces). By default, CLIQUE uses one-dimensional primitive subspaces, but in hubCLIQUE, we directly use two-dimensional primitive subspaces since this is the size of the projections used by the participants.

We customize the function that computes the density of points by adjusting the weights of each data point $x$ with the information collected from the solutions of Colony B. More specifically, we tune the weight of a data point using the definition below.

**Point Frequency** Let $X$ be a multidimensional dataset and $x \in X$ a data point. We note the frequency the players select a point $x$ in a two-dimensional cluster as $f_{\text{clustered}}(x)$, and the number of times a point $x$ is proposed to a player as $N_{\text{appeared}}(x)$ (i.e. the number of times a point appears in a puzzle). Then, we define the weight of a point $x$ as

$$W(x) = \frac{f_{\text{clustered}}(x)}{N_{\text{appeared}}(x)}.$$

We also design a filter to eliminate answers with an obvious bias (from the human solutions). This following step is new and does not exist in CLIQUE. It has been implemented to improve the quality of the clusters.

**Average cluster size** During the development of Colony B, we observed that many players selected large clusters. We hypothesize that these players assumed that larger clusters receive a larger score, even though it is rather the opposite. To account for this bias towards large clusters, we compute the average clusters size (ACS) over all solutions for each player and use this information to remove outliers (e.g. the clusters with a size exceeding the average by at least twice the standard deviation).

Other functions could be defined to filter the quality of solutions from participants and we experimented with several ones. All things considered, the one used in this paper offered the best results we have been able to obtain so far.

### 4.1.2. Main loop for building high-dimensional clusters

Once the weights for all points $x$ are computed, we use the original strategy of CLIQUE [44] to identify candidate clusters of points in subspaces with increasing dimensionality. More precisely, we merge pairs of clusters lying in two distinct subspaces of dimensions $K$ but having $(K-1)$ common dimensions. The points in the intersection of the two $K$-dimension clusters are selected for creating a new cluster in the merged $(K+1)$ dimensional subspace.

Noticeably, the new $(K+1)$ subspace has at most the same number of points as those covered by dense areas preserved from the two merged $K$ dimensional subspaces. It follows that a low coverage of data points is a frequent phenomenon, which can make the interpretation of results challenging. For instance, the original CLIQUE algorithm uses a minimal description length (MDL); pruning this searches for the optimal split based on the encoded coverage and its code length. Here, we choose to not prune too drastically subspaces with low coverage as they could belong to a larger (high-dimensional) solution.

Here, we first sort each single cluster based on (i) the number of dimensions and (ii) the size of the cluster in decreasing order. Then, we traverse this sorted list to identify subspaces with maximal overlaps and merge them.

Formally, let $S$ be a list of clusters from Colony B sorted by size and $s_i$ an element of $S$. The algorithm has two phases shown in algorithms 1 and 2. The first phase (algorithm 1) builds single multidimensional cluster candidates from two-dimensional annotations collected by Colony B. Iteratively, it computes the overlap between clusters $s_i \in S$ and $s_j \in S$ ($s_j \neq s_i$) in decreasing order of the cluster candidates in $S$ and removes (when necessary) overlapping elements. Then, the second phase (algorithm 2) scans the list of single multidimensional clusters sorted by size of the subspace and cardinality of the clusters. It assigns all data points in the cluster with highest priority in this queue. The remaining set of cluster candidates is sorted after every iteration based on the number of dimensions and number of conserved points. The algorithm terminates when the procedure passes over the entire set of candidate clusters.

hubCLIQUE outputs a vector assigning a cluster label to each point of the dataset. Eventually, some points remain without any cluster assignment (a more detailed description of the algorithm is provided in the electronic supplementary material).

**Algorithm 1.** hubCLIQUE (Phase 1) aggregates the crowdsourced 2D annotations from Colony B into single multidimensional clusters.

---

 **Input:** List $S$ of 2D clusters from Colony B sorted by size
 **Output:** List $C$ of single multi-dimensional clusters
1 $C \leftarrow \emptyset$
2 **for** $i \leftarrow 0$ *to* $|S|-1$ **do**
3 **if** $s_i$ *is not used* **then**
4 mark $s_i$ as used
5 $candidate \leftarrow s_i$
6 **for** $j \leftarrow i+1$ *to* $|S|-1$ **do**
7 **if** $s_j$ *is not used* **and** $dim(s_i \cap s_j) > 0$ **then**
8 **if** $\frac{|s_j \cap s_i|}{|s_j|} \geq 0.8$ **then**
9 mark $s_j$ as used
10 $candidate \leftarrow candidate \cup s_j$
11 Remove data points from $s_i$ occurring in less than half of clusters $s_j$
12 $C \leftarrow C \cup candidate$
13 **return** $C$

**Algorithm 2.** hubCLIQUE (Phase 2) processes a list of single multidimensional clusters to generate an assignment of all data points to a single cluster. Note that some data points can remain unassigned.

---

 **Input:** List $C$ of single multi-dimensional clusters
 **Output:** Partition $V$ of data point into multi-dimensional clusters
1 **for** $i \leftarrow 0$ *to* $|C| - 1$ **do**
2 **Sort** clusters in C[i+1 :] by:
3 **1:** Decreasing number of dimensions
4 **2:** Decreasing ratio of point only in $c_i$
5 **3:** Decreasing size of the cluster
6 **for** $j \leftarrow i + 1$ *to* $|C| - 1$ **do**
7 $c_j \leftarrow c_j \setminus c_i$
8 $V \leftarrow$ Empty vector of size $N$
9 **for** $i \leftarrow 0$ *to* $|C| - 1$ **do**
10 **for** $x \in c_i$ **do**
11 V[x] $\leftarrow$ i
12 **return** $V$

---

## 4.2. CloCworks

In this section, we propose a different approach to predicting clusters using the annotation collected from Colony B. We call this algorithm CloCworks. In contrast to hubCLIQUE, CloCworks aims to detect the occurrence of groups of consistent answers rather than to exploit the density of data.

CloCworks models the data collected from Colony B as a network and uses a community search algorithm to find pseudo-optimal partitions in this network. Since the problem is $\mathcal{NP}$-hard [45], we choose the Louvain community search algorithm [46] because of its performance (modularity score) and speed in comparison with other related algorithms [47] (see the electronic supplementary material for a graphical illustration of the algorithm).

### 4.2.1. Network construction

We build a network for every pair of dimensions used in the dataset analysed by Colony B. Each node of the network represents a data point, whereas the edges model the probability of two data points to be clustered together. Hence, the weight of the edges encodes the observed frequency of occurrence of a pair of points in a cluster.

Formally, let $P_k^{i,j}$ be a stage of the game for two dimensions $\{i, j\}$. The algorithm analyses all solutions for $P_k^{i,j}$ and computes the frequency of co-clustering two points together as the number of times the points are selected together over the number of times this stage was presented to a player. The results for all pairs of points are stored in a similarity matrix and we repeat this procedure for all stages in the $\{i, j\}$ subspace. Then, we average all the similarity matrices over all pairs of dimensions to produce a summary network that is processed in the next step of CloCworks. An illustration of the full process can be found in the electronic supplementary material.

### 4.2.2. Community detection and clustering

Once the summary network is assembled, we partition this data using the Louvain community search algorithm. All partitions of this network (aggregated for each pair of dimensions) are stored in a list $S$ of single clusters $s_i$. In contrast to the list of clusters used by hubCLIQUE that contains subspaces of different dimensionalities, CloCworks uses only two-dimensional clusters. Therefore, we modify the procedure used to aggregate the clusters in higher-dimensional spaces. In particular, we relax the thresholds used to merge intermediate clusters and allow for more diverse combinations of dimensions.

CloCworks can be seen as a variant of the correlation clustering approach applied to general weighted graphs [48]. However, instead of ± labelled edges with a weight denoting the confidence in labels, the proposed algorithm operates only with pairwise point similarities encoded as edge weights. Also due to very sparse dataset coverage by puzzles of Colony B, very few network clustering algorithms can be used. For example, algorithms operating with the clustering coefficient and transitivity [49] will perform poorly since the chance of finding triangles in the network observed

from Colony B is very low. Also, a spectral clustering [50] algorithm will generate a multitude of small clusters that in our case is not representative.

# 5. Experiments

In this section, we benchmark our two algorithms against standard clustering algorithms and the original CLIQUE algorithm. The simplicity of the methods tested (including ours), as well as their broad availability, allow a fairer and better interpretation of the results. We use a synthetic dataset and a real-world dataset to assess the accuracy of the proposed algorithm. Both datasets were played by Colony B players for two weeks in March 2019. The players were not preselected in any way, and only happened to be the ones actively playing during the period of trials. We did not make any particular announcement for the occasion. All the classic algorithms were tested with varying sets of parameters and only the best result for each algorithm was reported.

## 5.1. Synthetic dataset

First, we generate a synthetic dataset with high-density clusters in specific subspaces. Unlike the 'Synthetic data generation' procedure described in the original CLIQUE article [44], we avoid using predefined hyper-rectangles and their connectivity. Instead, we define the dimensionality (for simplicity, we choose six to comply with the requirements of our game system), the approximate size of a dataset, the number of clusters, and the ratio of additional noise. For each cluster, the generator randomly selects (with replacement) its dimensionality (and specific dimensions) from all the possible combinations of size in the range [2, 6], its size (number of points), the coordinates of the centre (mean), the shape, and the orientation (covariance). All these parameters are then used to draw random samples from a multivariate normal distribution. We restrict the minimum Manhattan distance between means of the clusters that share at least one dimension, as shown in the following equation:

$$\text{dist}_{\text{Man}}(\textit{mean}_1, \textit{mean}_2) > \text{dist}_{\text{Man}}(\mathbf{0}, \sqrt{\textit{cov}_1} + \sqrt{\textit{cov}_2}),$$

where $\textit{mean}_x$ and $\textit{cov}_x$ are the mean and covariance vectors for elements of cluster $x$ in the shared subspace, respectively.

For our experiment, we choose to generate a six-dimensional dataset with a size of approximately 2000 data points, which are randomly distributed over six clusters of various dimensionalities. Also, we added extra ('noise') points randomly distributed over the search space (5% of the dataset size). Table 1 shows generic information about the synthetic dataset.

To measure the accuracy of the different algorithms, we use the F1 score with micro averaging, which globally counts true positives, false positives, and false negatives rates to compute the average metric. This approach allows us to understand which clustering result was the closest one to the true labels. Importantly, this F1 score is only used *a posteriori* for evaluating the accuracy of the cluster predictions computed by our algorithms with the Colony B annotations. It is never used in the game or seen by the participants (the score seen by the participant is described in §3.4).

We compare the performance of hubCLIQUE and CloCworks using the collected human input of 25 players who submitted over 400 solutions during the two-week period (3–17 March 2019), along with the most popular clustering algorithms described below.

We include in the test three categories of algorithms [51]: (i) the original CLIQUE algorithm; (ii) algorithms that require a prespecified number of clusters (KMeans, Affinity Propagation (AP), Hierarchical clustering using Ward's minimum variance method (Ward), Gaussian Mixture Models (GMM)); and (iii) those that do not (DBSCAN, MeanShift).

Note that the GMM algorithm might converge to arbitrarily incorrect local maxima even in perfect conditions [52]. Therefore, we report the mean F1 score for 1000 runs instead (with standard deviation).

Since our hubCLIQUE algorithm does not require the number of clusters as a parameter, for the second category we estimate the number of clusters using the elbow method (which gives four as a result). We also tested them with the correct number of clusters (six), but the results were very similar and sometimes even worse (not shown).

Since this dataset consists of clusters in both high- (the top three clusters in table 1) and low- (the bottom three clusters in table 1) dimensional subspaces, we separately report the accuracy for both groups independently as well as for the whole dataset. All the results are shown in figure 3.

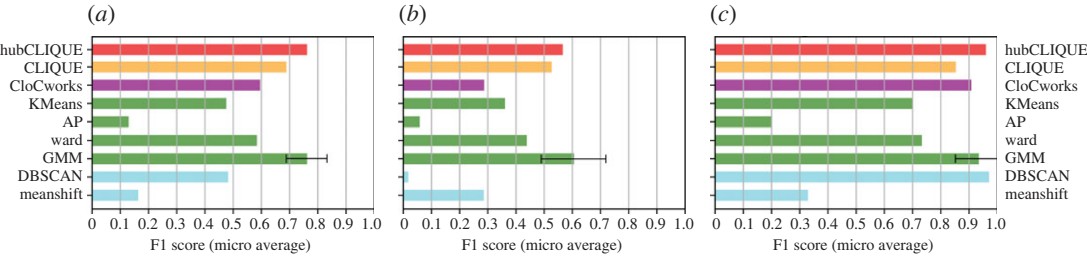

**Figure 3.** Performance comparison of the human-based algorithms with automated clustering approaches applied to the synthetic dataset for (*a*) all clusters; (*b*) low-dimensional clusters; and (*c*) high-dimensional clusters. The colour scheme is used to separate groups of algorithms: hubCLIQUE (red), automated CLIQUE (orange), CloCworks (purple), algorithms that require a known number of clusters/components (green), and algorithms that do not require it (cyan). For GMM, the average value over 1000 runs is reported. For algorithms requiring a known number of clusters, we report their performance with the number of clusters estimated by the elbow heuristic (four).

**Table 1.** Synthetic dataset information.

| subspace | cluster ID | size |
|---|---|---|
| {0, 3, 4, 5} | 0 | 338 |
| {1, 2, 4, 5} | 1 | 328 |
| {0, 1, 3, 4, 5} | 2 | 334 |
| {0, 1} | 3 | 340 |
| {0, 4} | 4 | 335 |
| {0, 4} | 5 | 339 |

First of all, the results clearly show that hubCLIQUE performs better than the original CLIQUE algorithm on both types of clusters, which demonstrates that the human input is positively contributing to the cluster identification process. More specifically, figure 3*c* demonstrates that the underlying technique in the hubCLIQUE algorithm does benefit from the solutions collected from various projections of the dataset and allows us to deal with multidimensional clusters. Point weights obtained from the human input help to eliminate points that otherwise get selected by the original CLIQUE algorithm, which, in turn, significantly improves the results (an increase of 12.5% in the micro F1 score).

Interestingly, DBSCAN shows good performance for high-dimensional clusters, whereas it performs poorly for low-dimensional ones. Indeed, the detection of those low-dimensional clusters is a very challenging task for all clustering algorithms because the clusters are embedded in subspaces with much fewer dimensions. In this context, most dimensions become irrelevant and coordinates in these subspaces produce a background noise that flattens the signal captured by the distance metrics. In addition, it is worth noting that two of the low-dimensional clusters (i.e. clusters 4 and 5) may be harder to distinguish as they are lying in the same subspace (table 1) and possibly occupy a similar region (see electronic supplementary material). In our experiments, DBSCAN appears to be particularly affected by these phenomena. This is also the case for *k*-means. By contrast, hubCLIQUE shows one of the best results among the tested algorithms.

When considering all clusters, our experiment shows that both hubCLIQUE and GMM (on average) perform the best. However, GMM does not yield stable results (recall that we report the average score over 1000 runs). Although the results of GMM can sometimes be good, its accuracy can significantly degrade on some runs for both for low- and high-dimensional clusters (as shown by the standard deviation).

Although CloCworks significantly differs from the other algorithms and considers human input as the main and only source of information about the dataset, it performs on par with the best performers in this test for high-dimensional clusters. For low-dimensional clusters, in contrast to hubCLIQUE, it produces less accurate results. Nevertheless, the algorithm detects the signal while suffering from the sparsity of the networks as well as the imperfection of the human solutions. Both of these aspects of the algorithm would need to be addressed in future work.

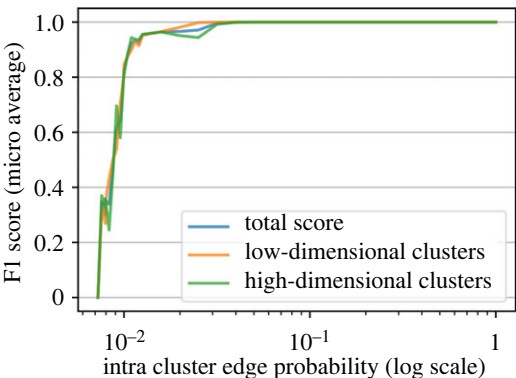

**Figure 4.** Analysis of the performance of CloCworks applied to a series of Colony B-simulated networks using the stochastic block models (SBM) network generator. CloCworks was applied to every set of networks.

## 5.2. Synthetic networks

Since CloCworks does not deal with the dataset directly, it is challenging to compare its accuracy with other clustering algorithms. Instead, we investigate its performance within a controlled environment. We generate multiple sets of Colony B-style networks corresponding to various configurations (different distributions of edges and their weights) using the stochastic block models (SBM) [53] network generator. Similarly, we consider $n$ vertices, which are split into $k$ communities of variable size. We connect nodes independently with probability $p_{i,j}$ where $i, j \in [k]$. Instead of generating equally weighted edges, we assign the weights $W_{i,j} \pm \sigma_{i,j}$ where $i, j \in [k]$ and $\sigma_{i,j}$ is the expected standard deviation.

Similarly to the previous experiment, for each set we build 15 networks with the same layout (as in table 1). For each network, the weight distribution follows the rule $0.8 \pm \sigma / 0.4 \pm \sigma / 0.1 \pm \sigma$ for intra-cluster, inter-cluster, and noise-related edges, respectively. We choose an expected standard deviation of $\sigma_{x,y} = 0.1$ for all three types of cluster relationship. The only variable is the probability of observing the intra-cluster edge $p_{intra}$, which varies in the range [0.005, 1]. Inter-cluster and noise-related edges are generated with probabilities $p_{inter} = 0.5 * p_{intra}$ and 0.2 (constant), respectively.

We also report the performance of CloCworks for high-dimensional clusters (Cluster ID {0, 1, 2} in table 1), low-dimensional clusters (Cluster ID {3, 4, 5} in table 1), and for the entire dataset (total score). The result of the experiment is shown in figure 4.

With $p_{intra} = 0.01$ we can get an F1 score of 0.88 for all three measures. This is expected since, in this case for an implicit cluster containing 300 points (nodes) on average, each node would be connected with only three other nodes. Reducing the intra-cluster probability causes the appearance of disjoint parts in the networks and therefore can result in a multitude of small false clusters. Such an accuracy implies that it should be possible to recover the majority of communities (clusters) from the datasets uploaded to Colony B given that the game puzzles cover the datasets in an extensive manner.

## 5.3. Voice recognition dataset

To examine the performance of the proposed algorithms on the real-world dataset, we apply our techniques to a voice recognition dataset (VRD) [54], which consists of over 20 measures of either male or female voice parameters. However, in order to keep the task tractable with our resources, we reduced the dimensionality of this dataset to six dimensions only.

Hence, as a preprocessing step, we use a random forest classifier (fitting 25 trees) on the VRD dataset with predefined *male/female* labels as a target to identify six features that can be used in our system. The extracted features (*meanfun*, *IQR*, *Q25*, *sd*, *sp.ent*, and *sfm*) are then used to create Colony B puzzles. The VRD-related puzzles were played by 75 volunteers with over 700 solutions submitted during a two-week period (March 2019).

As we did earlier, we benchmark both our approaches against CLIQUE, KMeans, Ward, GMM, and DBSCAN (figure 5). We ran AP and MeanShift, but omitted them due to poor results, similar to their low average performance in the previous experiment. For algorithms that require a known number of clusters (green category), we report an algorithm's performance with the number of clusters estimated with the

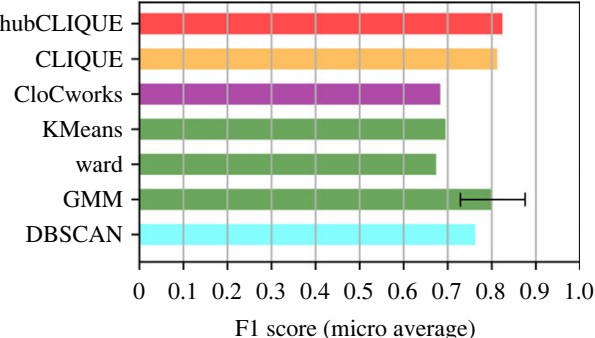

**Figure 5.** Performance comparison of the human-based algorithms with automated clustering approaches applied to the voice recognition dataset (VRD). The figure notations and chosen colour scheme are described in figure 3.

elbow method (which resulted in three, although the correct number of clusters was two). We also used the elbow method to find the required DBSCAN parameters (*eps* and *min_samples*).

Our results show that hubCLIQUE in general performs slightly better than the CLIQUE algorithm and the GMM algorithm (on average). We can observe a significant decrease in performance for Ward and KMeans with the estimated number of clusters. It is interesting that GMM performed well even with the incorrect number of clusters as an input parameter. The reason is that the actual two clusters were generally well identified and the outliers were mostly distributed in the extra (third) cluster.

By contrast, the performance of DBSCAN is average, mostly because it identified a single major cluster with a lot of outliers. It shows that the density-based clustering approach, which considers a high-dimensional dataset as is, might present a challenge with finding parameters that will lead to a more or less realistic clustering result, which is not simply one big cluster.

Finally, CloCworks does not perform better than Ward and KMeans on the real dataset. This can be explained in part by the fact that clusters are not as well defined in this dataset as in the synthetic dataset, and the CloCworks algorithm relies a lot more on the connections provided from the human solutions (without really considering the data itself, as in hubCLIQUE).

## 6. Discussion

In this paper, we implement a mobile human-computing game, Colony B, to collect human input for guiding the clustering of multidimensional datasets, and introduce two semi-automated multidimensional data clustering approaches called hubCLIQUE and CloCworks that leverage the annotations collected from Colony B to compute multidimensional clusters.

While hubCLIQUE uses collected human annotations as assisting information for a density correction within the seminal CLIQUE subspace clustering algorithm, CloCworks applies a community search strategy to find a solution entirely based on aggregated human solutions.

To evaluate the performance of our methods, we have conducted two experiments using a synthetic dataset and a real-world dataset that involved a total of 100 participants who submitted over 1100 solutions. We have benchmarked our proposed methods against some of the most popular automated clustering algorithms.

On both datasets, we have showed that hubCLIQUE outperformed its competitors in identifying both high- and low-dimensional clusters. The CloCworks approach has also showed competitive results for high-dimensional clusters but appeared less efficient with low-dimensional clusters and the real-world dataset. Nevertheless, it still performed on par with other traditional methods. We also note that CloCworks benefits more from human input when the dataset contains less structural uncertainty (e.g. the synthetic dataset).

As stated before, these results were observed using the data accumulated during a two-week period with Colony B. It illustrates that it is possible to analyse a dataset using players with no knowledge of the underlying problem in a relatively short amount of time given that (i) the game rules are transparent; and (ii) the game preserves a natural game progression balancing the challenge level with the skill level of a player.

## 6.1. Limitations

Although our experiments demonstrate an improvement of the predictions, the magnitude of this improvement remains modest. But it is reassuring and non-obvious that adding crowdsourced annotations does improve the performances. To have a practical long-term impact and broad adoption of this approach, more significant improvements are required. In particular, our algorithms hubCLIQUE and CloCworks could be refined, but other strategies to leverage the crowdsourced annotations should be explored as well.

Interestingly, we have tested CloCworks by applying it to a set of Colony B-simulated networks generated using an SBM network generator. We have examined its performance on a wide variety of configurations of the networks (from sparse to fully connected) and identified that the algorithm starts to misclassify data points in very sparse networks (probability of observing intra-cluster edge $p_{intra} < 0.01$). This result confirms that the puzzles' coverage of the data points in Colony B can be rather sparse, but we could not characterize the minimal resolution required for guaranteeing robust performances yet. We also found that CloCworks is sensitive to the outliers and can be confounded by a misleading human solution. At this stage, its performance and benefits are debatable, but we believe it still offers an interesting comparison and a different perspective for aggregating crowdsourced human annotations.

Even though our methods have been designed to analyse high-dimensional datasets, the experiments presented in this article are limited to datasets with a maximum of six dimensions. We made this choice in order to enable a limited crowd to process all two-dimensional projections of the main dataset. While we believe our results provide us with enough evidence of the potential of this approach, we acknowledge that more extensive experiments are warranted to fully establish this technology.

## 6.2. Scalability and sustainability

The data used in this work have been obtained using a mobile application published by our research team on the Apple App Store and Google Play. Since its release in Google Play and the Apple App Store in August 2016, our application has been downloaded and played by more than 10 000 mobile users. This setting has enabled us to engage 100 participants in a short period of time and collect enough solutions for this preliminary study. During the period in which we conducted these experiments, the average number of solutions collected per day was approaching 50. Interestingly, this rate remained steady over time. In this work, we aimed to collect enough data to guarantee that a data point appears in approximately three stages of different puzzles. But the number of puzzles required may vary with different parameters of the game, such as the number of puzzles or the number of data points used in each stage. For instance, the VRD dataset has about 2000 data points ($N$) for which we randomly generated 50 puzzles ($P$) made of 100 data points each ($n$). This results on average to $P \times n/N = (50 \times 1000)/2000 = 2.5$ puzzles per data point, which is consistent with our initial objectives. Of course, these values will need to be adjusted for different datasets and numbers of participants.

Without a massive professional promotional campaign, it is unlikely that this strategy can scale up and generate enough throughput to rapidly process large datasets on demand. An option could be to use crowdsourcing services such as Amazon Mechanical Turk or Appen, but this approach would be costly and still limited to crowds of tens of thousands of workers with specific dynamics [55]. By contrast, our long-term objective is to leverage the scientific motivation and game mechanics to integrate this task in a popular video game and reach out to larger crowds [56]. In fact, this study aimed to provide us preliminary data for a larger project with CCP games, which already conducted similar experiences in Eve Online [57]. More details about the current initiative are available in the press release of the new phase of this project [58].

## 6.3. Conclusion

This work introduces the first step towards a general framework for harnessing the collective wisdom of crowds of participants to improve the performance of algorithms on hard computational problems. Despite the stochastic nature of the annotations collected, it is worth noting that our methods seem to produce robust results. The latter could be counterintuitive since we are leveraging human perception, but our assumption is precisely that larger crowds result in more stable agreements. The quantification of this phenomenon and analysis of the limitation of this hypothesis could be addressed in future work.

It is also important to emphasize that using crowdsourced data does not automatically result in improved performance. Indeed, the integration of information generated by non-experts can also create a noise that deteriorates the results of automated methods. In this context, our results are promising and we believe that even a small improvement is significant. The complexity of deploying a complete crowdsourcing framework suggests that this technique may not be ready for broad adoption yet. However, we believe it unveils promising research directions.

Finally, the use of a casual puzzle game embedded in a mobile application appears to be a promising approach to deploy this technology and gather participants. Further work on the gamification of this clustering task can broaden the audience and the performance of this platform.

Ethics. We accepted all players without a selection process and gave them a choice to register with email or start playing right away as a guest user. In the case of email registration, we did not store any personal information. We explicitly guaranteed the confidentiality of collected information (i.e. clusters) and stated that it would only be used to enable a user to reset the password or to send important updates to the user. The privacy policy can be found on the Colony B server website.

Data accessibility. Colony B is available on Google Play (https://play.google.com/store/apps/details?id=pontax.cs.mcgill.ca) and the Apple App Store (https://apps.apple.com/ca/app/colony-b/id988892383). The voice recognition dataset is available on the Kaggle website at https://www.kaggle.com/rohankale/voice-recognition. The source code of our algorithms and all the data collected and used in this study are available at http://csb.cs.mcgill.ca/colonyb-data or at https://doi.org/10.5061/dryad.qv9s4mwh4 [59].

Additional material is provided in the electronic supplementary material [60].

Authors' contributions. A.B.: conceptualization, data curation, formal analysis, methodology, software, validation, visualization, writing—original draft; C.D.: software, visualization; O.T.-S: data curation, formal analysis, methodology, writing—review and editing; J.W.: conceptualization, formal analysis, funding acquisition, methodology, supervision, validation, writing—review and editing.

All authors gave final approval for publication and agreed to be held accountable for the work performed therein.

Conflict of interest declaration. We declare we have no competing interests.

Funding. This work was supported by Genome Canada, Genome Québec, and Canadian Institutes of Health Research through the Bioinformatics and Computational Biology competition.

Acknowledgements. The authors would like to thank all the Colony B players who made this study possible.

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
