## [Peer Review File · Royal Society Open Science]

Review History

RSOS-201321.R0 (Original submission)

Review form: Reviewer 1

Is the manuscript scientifically sound in its present form?

Yes

Are the interpretations and conclusions justified by the results?

Yes

Is the language acceptable?

Yes

Do you have any ethical concerns with this paper?

No

Have you any concerns about statistical analyses in this paper?

No

Recommendation?

Accept with minor revision (please list in comments)

Comments to the Author(s)

Authors propose two clustering algorithms that use human supervision through gamification. This idea is very innovative, and results have shown its effectiveness.

In the following some minor comments:

- authors state "In fact, the presence (or absence) of a cluster ultimately results from an agreement between multiple individuals, and preferentially data analysis experts." Could you please add a reference for this sentence, or better justify it? Usually automatic evaluation measures are used for clustering validation. It is the first time I heard that experts, or groups of experts are involved in the clustering evaluation. This could happen, after the analysis, but this is not crucial for the clustering process.
- It is not clear why and how F1 score is used. Unless you refer to a different measure, F1 score is a supervised measure. I think that unsupervised measures should be used to evaluate the clustering results. In paragraph (d) you wrote that "three clustering validation indices: (i) Silhouette, (ii) SDbw, and (iii) Dunn" are used. Thus, it is not clear if you have used these three measures or F1 score. Could you please better explain this point? However, if a supervised measure has been used to evaluate clustering results, I suggest to replace it with an unsupervised one.
- figure 3 shows that k-means algorithm performs better with high dimensional data, than with low ones. This is very weird, since this algorithm is affected by the 'curse of dimensionality' that you mentioned before. Could you please explain this result? The same algorithm obtains quite good results also with the real data. This is also quite surprising.
- Could you please better detail the real data you used? Particularly, since the high dimensionality could affect the results, you should provide the dimensions of the dataset

Review form: Reviewer 2

Is the manuscript scientifically sound in its present form?

No

Are the interpretations and conclusions justified by the results?

No

Is the language acceptable?

Yes

Do you have any ethical concerns with this paper?

No

Have you any concerns about statistical analyses in this paper?

Yes

Recommendation?

Reject

Comments to the Author(s)

Strengths:

- * The paper tackles an old problem with a clever approach that uses human supervision.
- * The crowd is involved via gamification; the case of citizen science is not applicable.
- * The algorithm hubClique is an intuitive adaptation of the Clique algorithm. It uses the crowd's input as initial cluster points and builds bigger subspaces using consensus-based heuristics.

Areas of improvements:

- * It is not clear if the incentive is sustainable. Having tried the app, the parallel with the biome data is clearly artificial.
- * Given the limitations of crowdsourcing-based methods, consider rooting the work in a specific use case.
- * The app allows selecting only 1 cluster per dimension pair. Is that the normal workflow?
- * The text argues for the limitation of clustering methods in high-dimensional data, yet the synthetic experiments are limited to six dimensions and six clusters.
- * A discussion on the complexity of the algorithms is missing. Especially with human factors, the setup might not be scalable.
- * It is better to switch to MTurk to obtain a larger number of labels and perform a more comprehensive evaluation.
- * Algorithm 1 requires rewriting. Multiple variables and data structures are introduced without explanation. Also, it is the central piece of the work given the performance of hubClique. It should be integrated into the paper. Specifically, the text in section 4 is not sufficient to properly understand the method.
- * The benefits of ColCworks are not clear.
- * The performance of DBScan on low dimensional clusters is suspect. Please provide a visualization.
- * Consider utilizing the same data generation of Clique (hyper rectangle) to understand the added value of hubClique.

Decision letter (RSOS-201321.R0)

Dear Dr Waldispuhl

The Editors assigned to your paper RSOS-201321 "Human-supervised Clustering of Multidimensional Data using Crowdsourcing" have made a decision based on their reading of the paper and any comments received from reviewers.

Regrettably, in view of the reports received, the manuscript has been rejected in its current form. However, a new manuscript may be submitted which takes into consideration these comments.

We invite you to respond to the comments supplied below and prepare a resubmission of your manuscript. Below the referees' and Editors' comments (where applicable) we provide additional requirements. We provide guidance below to help you prepare your revision.

Please note that resubmitting your manuscript does not guarantee eventual acceptance, and we do not generally allow multiple rounds of revision and resubmission, so we urge you to make every effort to fully address all of the comments at this stage. If deemed necessary by the Editors, your manuscript will be sent back to one or more of the original reviewers for assessment. If the original reviewers are not available, we may invite new reviewers.

Please resubmit your revised manuscript and required files (see below) no later than 13-Dec-2021. Note: the ScholarOne system will 'lock' if resubmission is attempted on or after this deadline. If you do not think you will be able to meet this deadline, please contact the editorial office immediately.

Please note article processing charges apply to papers accepted for publication in Royal Society Open Science (<https://royalsocietypublishing.org/rsos/charges>). Charges will also apply to papers transferred to the journal from other Royal Society Publishing journals, as well as papers submitted as part of our collaboration with the Royal Society of Chemistry (<https://royalsocietypublishing.org/rsos/chemistry>). Fee waivers are available but must be requested when you submit your manuscript (<https://royalsocietypublishing.org/rsos/waivers>).

Thank you for submitting your manuscript to Royal Society Open Science and we look forward to receiving your resubmission. If you have any questions at all, please do not hesitate to get in touch.

on behalf of Professor Mirella Lapata (Associate Editor) and Marta Kwiatkowska (Subject Editor)
openscience@royalsociety.org

Reviewer comments to Author:

Reviewer: 1

Comments to the Author(s)

Authors propose two clustering algorithms that use human supervision through gamification. This idea is very innovative, and results have shown its effectiveness.

In the following some minor comments:

- authors state "In fact, the presence (or absence) of a cluster ultimately results from an agreement between multiple individuals, and preferentially data analysis experts." Could you please add a reference for this sentence, or better justify it? Usually automatic evaluation measures are used for clustering validation. It is the first time I heard that experts, or groups of experts are involved in the clustering evaluation. This could happen, after the analysis, but this is not crucial for the clustering process.
- It is not clear why and how F1 score is used. Unless you refer to a different measure, F1 score is a supervised measure. I think that unsupervised measures should be used to evaluate the clustering results. In paragraph (d) you wrote that "three clustering validation indices: (i) Silhouette, (ii) SDbw, and (iii) Dunn" are used. Thus, it is not clear if you have used these three measures or F1 score. Could you please better explain this point? However, if a supervised measure has been used to evaluate clustering results, I suggest to replace it with an unsupervised one.
- figure 3 shows that k-means algorithm performs better with high dimensional data, than with low ones. This is very weird, since this algorithm is affected by the 'curse of dimensionality' that you mentioned before. Could you please explain this result? The same algorithm obtains quite good results also with the real data. This is also quite surprising.
- Could you please better detail the real data you used? Particularly, since the high dimensionality could affect the results, you should provide the dimensions of the dataset

Reviewer: 2

Comments to the Author(s)

Strengths:

- * The paper tackles an old problem with a clever approach that uses human supervision.
- * The crowd is involved via gamification; the case of citizen science is not applicable.
- * The algorithm hubClique is an intuitive adaptation of the Clique algorithm. It uses the crowd's input as initial cluster points and builds bigger subspaces using consensus-based heuristics.

Areas of improvements:

- * It is not clear if the incentive is sustainable. Having tried the app, the parallel with the biome data is clearly artificial.
- * Given the limitations of crowdsourcing-based methods, consider rooting the work in a specific use case.
- * The app allows selecting only 1 cluster per dimension pair. Is that the normal workflow?
- * The text argues for the limitation of clustering methods in high-dimensional data, yet the synthetic experiments are limited to six dimensions and six clusters.
- * A discussion on the complexity of the algorithms is missing. Especially with human factors, the setup might not be scalable.
- * It is better to switch to MTurk to obtain a larger number of labels and perform a more comprehensive evaluation.
- * Algorithm 1 requires rewriting. Multiple variables and data structures are introduced without explanation. Also, it is the central piece of the work given the performance of hubClique. It should be integrated into the paper. Specifically, the text in section 4 is not sufficient to properly understand the method.
- * The benefits of ColCworks are not clear.
- * The performance of DBScan on low dimensional clusters is suspect. Please provide a visualization.
- * Consider utilizing the same data generation of Clique (hyper rectangle) to understand the added value of hubClique.

===PREPARING YOUR MANUSCRIPT===

===PREPARING YOUR REVISION IN SCHOLARONE===

Author's Response to Decision Letter for (RSOS-201321.R0)

See Appendix A.

RSOS-211189.R0

Review form: Reviewer 3 (Firas Khatib)

Is the manuscript scientifically sound in its present form?

Yes

Are the interpretations and conclusions justified by the results?

Yes

Is the language acceptable?

Yes

Do you have any ethical concerns with this paper?

No

Have you any concerns about statistical analyses in this paper?

No

Recommendation?

Accept as is

Comments to the Author(s)

This is an exciting proof-of-concept paper that will interest many in the field!

I had some questions:

- 1) You mentioned "100 participants who submitted over 1100 solutions" over a 2-week period, and I believe that 700 of those 1100 solutions were VRD related puzzles played by 75 participants. Who were those participants in terms of ColonyB? Was it anyone who happened to play during those 2 weeks?

Was it only those who reached a specific status on the leaderboard, or had played at least X number of games beforehand, or played ColonyB for a minimum number of days?

2) When was this 2-week period and was it announced in-game that these puzzles would be any different than the regular ones, or on the contrary: did you intentionally not say anything?

3) How many participants have played ColonyB since it was posted on both app stores?

I ask because 100 participants sounds very low when reading the paper, but might be understandable based on my Question #1. It might help to mention that thousands of players have downloaded and tried ColonyB.

3) "We intentionally skipped AP and MeanShift due to their low average performance in the previous experiment."

I understand this, but did you actually run these two (and just decided not to plot the useless results) or did you never run them at all?

If it is quick to run them, it might be better to be able to state: "We ran AP and MeanShift, but omitted them from Figure X due to poor results, similar to their low average performance in the previous experiment."

4) "During the development of Colony B, we observed that many players selected large clusters. We hypothesize that these players assumed that larger clusters receive a larger score, even though it is rather the opposite."

As you observed this during ColonyB's development, did you change the instructions in the tutorial to address this for future puzzles (such as EVE online)?

5) Lastly, Figure 2 is very clear, but I have a suggestion: Panel D has a yellow circle (remaining from the previous screen) which is distracting... and is even faintly visible on Panel E.

Review form: Reviewer 4

Is the manuscript scientifically sound in its present form?

Yes

Are the interpretations and conclusions justified by the results?

Yes

Is the language acceptable?

Yes

Do you have any ethical concerns with this paper?

No

Have you any concerns about statistical analyses in this paper?

No

Recommendation?

Major revision is needed (please make suggestions in comments)

Comments to the Author(s)

This paper revolves around implementing two algorithms (named hubCLIQUE and CloCworks) for clustering multidimensional data. It has been claimed that these algorithms are novel.

However, the first method is based on the classic CLIQUE algorithm (tailored to be used in the concerned context). The second method uses a Louvain community detection algorithm.

1. The contributions of this paper are not apparent. The introduction of the paper is not sufficient to provide an apt summary of the problem to be solved and highlight the novelty of the work. A better explanation of the goal of the data analysis should be provided.
2. In the introduction and related work, most literature are old. It is suggested to cover more recent works to show this area is active and absorbing. Some are suggested here: "Clustering structure analysis in time-series data with density-based clusterability measure," "Accelerated Two-Stage Particle Swarm Optimization for Clustering Not-Well-Separated Data," "Extracting Significant Mobile Phone Interaction Patterns based on Community Structures," "A Novel Rolling Bearing Vibration Impulsive Signals Detection Approach Based on Dictionary Learning."
3. The technical content is not well described.
4. The study has not explained well why one should consider the implemented methods over other alternatives, or what other alternatives are available. Some conventional machine learning methods have been used in this work for comparison. But they treat community detection as a problem of verifying the accuracy of clusters on large graphs. They generate a network consisting of a few dimensions to reconstruct the original network. However, this type of representation to a low dimensional space is linear. The fact that the real-world networks include nonlinear structures makes the traditional strategies less useful. As the scale of networks increases in the real world, more robust and efficient techniques are required to achieve high performance.
5. Regarding the result section, the authors should notice that a separate dimensionality reduction phase is needed for using methods such as DBSCAN. It is obvious that without performing such a phase, the accuracy of traditional methods would degrade.
6. This work uses a bottom-up approach. These methods use the local structures and try to expand them to form clusters. They usually fail to detect tiny clusters or non-convex ones. For community detection using such methods, the initial local structures do not capture small communities from scratch, and the technique fails to incorporate node members of a community. The authors should also add more discussion regarding their network topology and node characteristics. Given the irregular structure of networks, more advanced methods based on deep learning have been proposed. The authors are invited to compare their implemented methods with such state-of-the-art approaches.
7. If the authors aim to propose novel clustering methods, they should discuss the novelty of their algorithms and compare them with state-of-the-art methods over standard datasets.
8. It will be better if the computation time can be provided for the proposed algorithm and its competitors in experiments since the efficiency also matters. More statistics data (e.g. MSE) should be provided to support the point "GMM does not yield stable results (recall that we report the average score over 1000 runs)" as GMM performs satisfactorily in most cases.

Decision letter (RSOS-211189.R0)

Dear Dr Waldispuhl

The Editors assigned to your paper RSOS-211189 "Human-supervised Clustering of Multidimensional Data using Crowdsourcing" have now received comments from reviewers and

would like you to revise the paper in accordance with the reviewer comments and any comments from the Editors. Please note this decision does not guarantee eventual acceptance.

Please submit your revised manuscript and required files (see below) no later than 21 days from today's (ie 08-Dec-2021) date. Note: the ScholarOne system will 'lock' if submission of the revision is attempted 21 or more days after the deadline. If you do not think you will be able to meet this deadline please contact the editorial office immediately.

on behalf of Professor Mirella Lapata (Associate Editor) and Marta Kwiatkowska (Subject Editor)
openscience@royalsociety.org

Reviewer comments to Author:

Reviewer: 3

Comments to the Author(s)

This is an exciting proof-of-concept paper that will interest many in the field!

I had some questions:

1) You mentioned "100 participants who submitted over 1100 solutions" over a 2-week period, and I believe that 700 of those 1100 solutions were VRD related puzzles played by 75 participants. Who were those participants in terms of ColonyB?

Was it anyone who happened to play during those 2 weeks?

Was it only those who reached a specific status on the leaderboard, or had played at least X number of games beforehand, or played ColonyB for a minimum number of days?

2) When was this 2-week period and was it announced in-game that these puzzles would be any different than the regular ones, or on the contrary: did you intentionally not say anything?

3) How many participants have played ColonyB since it was posted on both app stores? I ask because 100 participants sounds very low when reading the paper, but might be understandable based on my Question #1. It might help to mention that thousands of players have downloaded and tried ColonyB.

3) "We intentionally skipped AP and MeanShift due to their low average performance in the previous experiment."

I understand this, but did you actually run these two (and just decided not to plot the useless results) or did you never run them at all?

If it is quick to run them, it might be better to be able to state: "We ran AP and MeanShift, but omitted them from Figure X due to poor results, similar to their low average performance in the previous experiment."

4) "During the development of Colony B, we observed that many players selected large clusters. We hypothesize that these players assumed that larger clusters receive a larger score, even though it is rather the opposite."

As you observed this during ColonyB's development, did you change the instructions in the tutorial to address this for future puzzles (such as EVE online)?

5) Lastly, Figure 2 is very clear, but I have a suggestion: Panel D has a yellow circle (remaining from the previous screen) which is distracting... and is even faintly visible on Panel E.

Reviewer: 4

Comments to the Author(s)

This paper revolves around implementing two algorithms (named hubCLIQUE and CloCworks) for clustering multidimensional data. It has been claimed that these algorithms are novel.

However, the first method is based on the classic CLIQUE algorithm (tailored to be used in the concerned context). The second method uses a Louvain community detection algorithm.

1. The contributions of this paper are not apparent. The introduction of the paper is not sufficient to provide an apt summary of the problem to be solved and highlight the novelty of the work. A better explanation of the goal of the data analysis should be provided.

2. In the introduction and related work, most literature are old. It is suggested to cover more recent works to show this area is active and absorbing. Some are suggested here: "Clustering structure analysis in time-series data with density-based clusterability measure," "Accelerated Two-Stage Particle Swarm Optimization for Clustering Not-Well-Separated Data," "Extracting Significant Mobile Phone Interaction Patterns based on Community Structures," "A Novel Rolling Bearing Vibration Impulsive Signals Detection Approach Based on Dictionary Learning."

3. The technical content is not well described.

4. The study has not explained well why one should consider the implemented methods over other alternatives, or what other alternatives are available. Some conventional machine learning methods have been used in this work for comparison. But they treat community detection as a problem of verifying the accuracy of clusters on large graphs. They generate a network consisting of a few dimensions to reconstruct the original network. However, this type of representation to a low dimensional space is linear. The fact that the real-world networks include nonlinear structures makes the traditional strategies less useful. As the scale of networks increases in the real world, more robust and efficient techniques are required to achieve high performance.

5. Regarding the result section, the authors should notice that a separate dimensionality reduction phase is needed for using methods such as DBSCAN. It is obvious that without performing such a phase, the accuracy of traditional methods would degrade.

6. This work uses a bottom-up approach. These methods use the local structures and try to expand them to form clusters. They usually fail to detect tiny clusters or non-convex ones. For

community detection using such methods, the initial local structures do not capture small communities from scratch, and the technique fails to incorporate node members of a community. The authors should also add more discussion regarding their network topology and node characteristics. Given the irregular structure of networks, more advanced methods based on deep learning have been proposed. The authors are invited to compare their implemented methods with such state-of-the-art approaches.

7. If the authors aim to propose novel clustering methods, they should discuss the novelty of their algorithms and compare them with state-of-the-art methods over standard datasets.

8. It will be better if the computation time can be provided for the proposed algorithm and its competitors in experiments since the efficiency also matters. More statistics data (e.g. MSE) should be provided to support the point "GMM does not yield stable results (recall that we report the average score over 1000 runs)" as GMM performs satisfactorily in most cases.

===PREPARING YOUR MANUSCRIPT===

If you have been asked to revise the written English in your submission as a condition of publication, you must do so, and you are expected to provide evidence that you have received language editing support. The journal would prefer that you use a professional language editing service and provide a certificate of editing, but a signed letter from a colleague who is a fluent speaker of English is acceptable. Note the journal has arranged a number of discounts for authors using professional language editing services (<https://royalsociety.org/journals/authors/benefits/language-editing/>).

===PREPARING YOUR REVISION IN SCHOLARONE===

Author's Response to Decision Letter for (RSOS-211189.R0)

See Appendix B.

RSOS-211189.R1

Review form: Reviewer 3 (Firas Khatib)

Is the manuscript scientifically sound in its present form?

Yes

Are the interpretations and conclusions justified by the results?

Yes

Is the language acceptable?

Yes

Do you have any ethical concerns with this paper?

No

Have you any concerns about statistical analyses in this paper?

No

Recommendation?

Accept as is

Comments to the Author(s)

Thank you for your revisions!

Review form: Reviewer 4

Is the manuscript scientifically sound in its present form?

Yes

Are the interpretations and conclusions justified by the results?

Yes

Is the language acceptable?

Yes

Do you have any ethical concerns with this paper?

No

Have you any concerns about statistical analyses in this paper?

No

Recommendation?

Accept with minor revision (please list in comments)

Comments to the Author(s)

The author has done a good revision job and addressed my previous comments. In the literature review, please discuss the following recent studies: Residual-driven Fuzzy C-Means Clustering

for Image Segmentation; and Multi-Cluster Feature Selection Based on Isometric Mapping. Please pay attention to English errors, e.g., “The simplicity of the methods tested (including ours) as well as their broad availability allows” => “The simplicity of the methods tested (including ours) as well as their broad availability allow”. In Sec. 6, the authors are suggested to use present tense or complete one. Please try to avoid past tense unless some past time adverbs are used in a sentence.

Decision letter (RSOS-211189.R1)

Dear Dr Waldispuhl,

On behalf of the Editors, we are pleased to inform you that your Manuscript RSOS-211189.R1 "Human-supervised Clustering of Multidimensional Data using Crowdsourcing" has been accepted for publication in Royal Society Open Science subject to minor revision in accordance with the referees' reports. Please find the referees' comments along with any feedback from the Editors below my signature.

Please submit your revised manuscript and required files (see below) no later than 7 days from today's (ie 18-Mar-2022) date. Note: the ScholarOne system will 'lock' if submission of the revision is attempted 7 or more days after the deadline. If you do not think you will be able to meet this deadline please contact the editorial office immediately.

on behalf of Professor Mirella Lapata (Associate Editor) and Marta Kwiatkowska (Subject Editor)
openscience@royalsociety.org

Reviewer comments to Author:

Reviewer: 3

Comments to the Author(s)

Thank you for your revisions!

Reviewer: 4

Comments to the Author(s)

The author has done a good revision job and addressed my previous comments. In the literature review, please discuss the following recent studies: Residual-driven Fuzzy C-Means Clustering for Image Segmentation; and Multi-Cluster Feature Selection Based on Isometric Mapping. Please pay attention to English errors, e.g., “The simplicity of the methods tested (including ours) as well as their broad availability allows” => “The simplicity of the methods tested (including ours) as well as their broad availability allow”. In Sec. 6, the authors are suggested to use present tense or complete one. Please try to avoid past tense unless some past time adverbs are used in a sentence.

===PREPARING YOUR MANUSCRIPT===

one version should clearly identify all the changes that have been made (for instance, in coloured highlight, in bold text, or tracked changes);

===PREPARING YOUR REVISION IN SCHOLARONE===

-- If you are requesting an article processing charge waiver, you must select the relevant waiver option (if requesting a discretionary waiver, the form should have been uploaded, see 'File upload' above).

-- If you have uploaded any electronic supplementary (ESM) files, please ensure you follow the guidance at <https://royalsociety.org/journals/authors/author-guidelines/#supplementary-material> to include a suitable title and informative caption. An example of appropriate titling and captioning may be found at https://figshare.com/articles/Table_S2_from_Is_there_a_trade-off_between_peak_performance_and_performance_breadth_across_temperatures_for_aerobic_scope_in_teleost_fishes_/3843624.

Author's Response to Decision Letter for (RSOS-211189.R1)

See Appendix C.

Decision letter (RSOS-211189.R2)

Dear Dr Waldispuhl,

I am pleased to inform you that your manuscript entitled "Human-supervised Clustering of Multidimensional Data using Crowdsourcing" is now accepted for publication in Royal Society Open Science.

on behalf of Professor Mirella Lapata (Associate Editor) and Marta Kwiatkowska (Subject Editor)
openscience@royalsociety.org

Appendix A

Human-supervised Clustering of Multidimensional Data using Crowdsourcing

Alexander Butyaev, Chrisostomos Drogaris, Olivier Tremblay-Savard & Jérôme Waldispühl

First of all, we would like to thank the reviewers for the careful and thoughtful review of our manuscript and for the concrete and constructive comments. We sincerely appreciate the time invested in reviewing our manuscript and the quality of the remarks.

Comments by Reviewer 1

Authors propose two clustering algorithms that use human supervision through gamification. This idea is very innovative, and results have shown its effectiveness.

We thank the reviewer for this appreciation of our work. We answer below to the comments and highlight the changes made to our manuscript to address them.

In the following some minor comments:

- authors state "In fact, the presence (or absence) of a cluster ultimately results from an agreement between multiple individuals, and preferentially data analysis experts." Could you please add a reference for this sentence, or better justify it? Usually automatic evaluation measures are used for clustering validation. It is the first time I heard that experts, or groups of experts are involved in the clustering evaluation. This could happen, after the analysis, but this is not crucial for the clustering process.

We added references to two seminal books about clustering [3, 7] that begins with a tentative definition of clusters. In these book, the authors note that the "formal definition is not only difficult but may even be misplaced" [3]. The discussions also emphasize the importance of domain-knowledge into the interpretation and validation of the results(e.g., Figure 1.2 in [7]).

- It is not clear why and how F1 score is used. Unless you refer to a different measure, F1 score is a supervised measure. I think that unsupervised measures should be used to evaluate the clustering results. In paragraph (d) you wrote that "three clustering validation indices: (i) Silhouette, (ii) SDbw, and (iii) Dunn" are used. Thus, it is not clear if you have used these three measures or F1 score. Could you please better explain this point? However, if a supervised measure has been used to evaluate clustering results, I suggest to replace it with an unsupervised one.

We acknowledge that our benchmark was not properly described. We have true labels for the synthetic and real datasets, but only use this information to assess the quality of the clustering results *a posteriori*. In particular, the labels are not used to score the clusters made by the players within the game. There is thus two scoring systems that are detailed below.

[Scoring of the clusters in the game (as seen by the participants)]

We do not use the true labels (i.e., unsupervised). The score showed to the participants is a normalized number that is high if one of the Silhouette, SDbw, or Dunn measure suggests the occurrence of a cluster. Otherwise, the score showed in the game is low. The Silhouette, SDbw, and Dunn measures are used to estimate the internal cohesion of a cluster made by the participant, and our score is designed to be non-penalizing. It grants points to the participants if the annotations has the potential to be "good" candidate and only penalizes the solutions that are very unlikely to be good candidate according to any of the metrics we use. Importantly, this score was not used in the final clustering result calculations (HubCLIQUE, CloCworks, KMeans, etc).

[A posteriori evaluation of the clusters computed from crowdsourced solutions]

We use the true labels (i.e., supervised). Here, our objective is to assess the accuracy of the clusters obtained with our methods (i.e., HubCLIQUE, CloCworks) and classical algorithms (e.g., k-means, DBSCAN) w.r.t. the ground truth. To that purpose, we use the micro-averaged F1 score (micro F1 score) to estimate the accuracy of the prediction to match the true labels (i.e., each cluster is represented as a binary vector that stores the label of each data point). In particular, we chose to use micro averaged version of the F1 score because of the data set is imbalanced.

We clarified this in the subsection "Synthetic dataset" (Subsection 5(a)).

- figure 3 shows that k-means algorithm performs better with high dimensional data, than with low ones. This is very weird, since this algorithm is affected by the 'curse of dimensionality' that you mentioned before. Could you please explain this result? The same algorithm obtains quite good results also with the real data. This is also quite surprising.

This is an important observation that indeed deserves to be clarified. To answer this question, we need to refer to Table 5.1 that describes clusters in the synthetic data set. This table shows that our data set includes "high" dimensional (e.g., the cluster with ID=2 lies in a 5 dimensional space) and "low" dimensional clusters (e.g., the cluster with ID=3 lies in a 2 dimensional space). Our objective is solve the sub-space clustering problem, which consist in finding clusters embedded in any sub-spaces. Here, we start with an initial set with 6 dimensions. We ran k-means on the complete (6D) data set and provide it the number of clusters to predict. The low and high dimensional terms refer to the dimensionality of the sub-space in which the clusters are embedded, and all clustering tasks are performed on the highest dimensional space. It follows that the distances between data points in

low dimensional clusters are noisy with several dimensions not relevant for accurately estimating the cohesion/separation between points. In this context, it is easier for k-means to identify clusters in higher dimensional sub-spaces (e.g., cluster with ID=2) than in lower dimensional sub-spaces (e.g., cluster with ID=3). This issue has previously been discussed and addressed with customized versions of k-means [4, 5]. In addition, as illustrated in a new section of the supplementary material (See “Supplementary information about the synthetic data set”) it turns out that two low dimensional clusters (i.e., clusters 4 and 5) are embedded in the same subspace (See Table 1) and occupy similar regions of this subspace. This situation can potentially affect the performances of k-means.

Next, the clusters in the voice recognition data set are lying in higher dimensional space (i.e., all dimension are informative). This data should be compared to high dimensional clusters in the synthetic data set and the performance are consistent (i.e., the F1 score is around 0.7 in both cases). We now describe this phenomenon in the subsection “Synthetic dataset” of the revised manuscript (p. 11) and added a new section of the supplementary material (See “Supplementary information about the synthetic data set”).

- Could you please better detail the real data you used? Particularly, since the high dimensionality could affect the results, you should provide the dimensions of the dataset

We apologize for the lack of details provided in the first version of our manuscript. The full data set can be found at: <https://www.kaggle.com/rohankale/voice-recognition> (link available at the end of the manuscript). Our initial data set has 20 features (dimensions). Though, we reduced the dimensionality of the data set used in the paper by selecting 6 dimensions according to the procedure described below: *As a preprocessing step, we use a random forest classifier (fitting 25 trees) on Voice Recognition dataset with predefined male/female labels as a target to identify six features that can be used for the Colony B game. Extracted features (named: meanfun, IQR, Q25, sd, sp.ent, and sfm) are then used to create Colony B puzzles. VRD related puzzles were played by 75 volunteers with over 700 solutions submitted during a two-week period.*

It has been emphasized in the revised manuscript in the section “Voice Recognition Dataset”.

Comments by Reviewer 2

Comments to the Author(s)

Strengths:

- The paper tackles an old problem with a clever approach that uses human supervision.
- The crowd is involved via gamification; the case of citizen science is not applicable.
- The algorithm hubClique is an intuitive adaptation of the Clique algorithm. It uses the crowd’s input as initial cluster points and builds bigger subspaces using consensus-based heuristics.

We thank the reviewer for highlighting the main contributions of our method. We answer below the remarks and issues identified in our initial manuscript and hope our answer will be satisfying.

Areas of improvements:

- It is not clear if the incentive is sustainable. Having tried the app, the parallel with the biome data is clearly artificial.

We acknowledge this concern and agree that the microbiome contextualization is not central to this project. We chose to integrate it because the app is currently using data from the American gut project (<https://microsetta.ucsd.edu/about/american-gut-project/>) and serves in parallel to conduct science outreach activities.

Yet, we understand that the sustainability of this initiative is the main concern here, and we concede that this consideration was not properly motivated in our initial submission. In fact, this project served as a preliminary study for another major citizen science initiative called Project Discovery (<https://www.eveonline.com/discovery>). In the latter, we aim to cluster multi-dimensional data from flow cytometry studies. More importantly, the application is embedded in a massively multiplayer online role-playing game (MMORPG) that has a community of hundred of thousands of players and ensures the sustainability of our initiative [6]. Before launching this project, we had to conduct preliminary studies to study the gamification mechanisms, develop algorithms to aggregate crowdsourced data, and estimate the potential of the approach. This work reports a preliminary study that aims to build the foundation of future related contributions, but also (we hope) will offer supporting data and inspiration for other researchers working in this area.

A new subsection (“Scalability and Sustainability“) has been added to the Discussion to clarify our strategy and current development phase of this project.

- Given the limitations of crowdsourcing-based methods, consider rooting the work in a specific use case.

As explained in our previous answer, we indeed intend to customize following versions of this project to a specific use case: flow cytometry data analysis. However, since this is not the main objective of this paper, we only implicitly suggest this use case in our revised discussion.

- The app allows selecting only 1 cluster per dimension pair. Is that the normal workflow?

It was a design choice we made for this specific project. Yet, the concept is by no mean restricted to one cluster. In following iteration of this project, we allow the participants to draw multiple clusters (See the reference to Project Discovery above). Yet, in this specific preliminary study we chose to use the simpler settings in order to facilitate the presentation and analysis of the data before expanding to the next milestones.

- The text argues for the limitation of clustering methods in high-dimensional data, yet the synthetic experiments are limited to six dimensions and six clusters.

We acknowledge this limitation. We intentionally decided to keep the size of the data manageable to enable us to guarantee realistic objectives with the limited resources (i.e., participants) available. In particular, our objectives were to consider all pairs of dimensions and guarantee a reasonable coverage. This was clarified in the discussion of the revised manuscript (See last paragraph of “Limitations“).

- A discussion on the complexity of the algorithms is missing. Especially with human factors, the setup might not be scalable.

This is indeed an important discussion that was missing in our initial manuscript. We now briefly discuss the scalability of our approach w.r.t. the human participation in a new subsection “Scalability and Sustainability” of the Discussion. We did not include a discussion of the computational complexity of the algorithms (in particular hubClique) since they are mostly for loops and sorting procedures, and run very quickly in practice.

- It is better to switch to MTurk to obtain a larger number of labels and perform a more comprehensive evaluation.

Using MTurk is indeed an option to conduct additional experiments. Though, as explained above, we opted for a different strategy that consists in integrating our task in a popular commercial video game. This information was not mentioned in the initial version of our manuscript. Since this paper is rather an introduction and proof-of-concept of our approach, we would prefer to postpone a more comprehensive evaluation of this technology when more data (from a new project) will become available. This has been emphasized in the last paragraph of the subsections “Limitations“ and “Scalability and Sustainability” of the Discussion. Yet, we also added a new reference to a seminal study of MTurk [2] in case a reader would be interested to apply our technique in this context.

- Algorithm 1 requires rewriting. Multiple variables and data structures are introduced without explanation. Also, it is the central piece of the work given the performance of hubClique. It should be integrated into the paper. Specifically, the text in section 4 is not sufficient to properly understand the method.

We agree that Algorithm 1 was poorly presented. We revised the pseudocode available in the supplementary material and added a text to describe it. Moreover, we added to the main manuscript a simplified version of this pseudocode, which we hope will allow the readers to quickly understand the main principles of the algorithm.

- The benefits of CloCworks are not clear.

We agree with the observation made by the reviewer. We developed CloCworks but were not able to demonstrate its superiority over other methods. However, we think the methods are different and potentially interesting for the reader for a comparison with another data aggregation strategy. The methods could also be potentially improved in the future and offer better performance in different use cases. This has been clarified in the second paragraph of the subsection “Limitations“ in the Discussion of the revised manuscript.

- The performance of DBScan on low dimensional clusters is suspect. Please provide a visualization.

This remarks echoes a similar observation made by Reviewer #1 for the k-means algorithm. We agree that this phenomenon needs to be more discussed. In the context of synthetic dataset, we call

low dimensional clusters, clusters that were generated in a (much) smaller subspace. For example, clusters with Cluster ID in 3,4,5 are embedded in a 2D subspace while the full data set has 6 dimensions. It is thus more difficult to identify these low dimensional clusters because the majority of coordinates are not relevant and only add more noise.

Another important difficulty faced by all algorithms when predicting low dimensional clusters is that two of these clusters are not easily separable. We tried to illustrate this phenomenon on a t-SNE projection of the synthetic data set below, in which we observe that clusters 4 and 5 are overlapping.

We now discuss this phenomenon in the revised manuscript and added a new section showing this data and discussing it in the supplementary material (See “Supplementary information about the synthetic data set”).

- Consider utilizing the same data generation of Clique (hyper rectangle) to understand the added value of hubClique.

We acknowledge this was not clearly stated in the manuscript. All algorithms have been tested on a (synthetic) data set generated by an algorithm similar to the hyper rectangle algorithm described in Original Clique paper [1]. The only difference is that we define dimensionality (for simplicity we choose six to comply with requirements of Colony B game) and the approximate size of a dataset, number of clusters, and ratio of additional noise. This allows us to generate more randomized noisy clusters as opposed to predefined hyper rectangles and connectivity. The standard deviation is assigned randomly. So it forms hyper ellipsoids rather than hyper rectangles, but the idea of “creating subspace clusters” is preserved.

References

- [1] Rakesh Agrawal, Johannes Gehrke, Dimitrios Gunopulos, and Prabhakar Raghavan. Automatic subspace clustering of high dimensional data. *Data Mining and Knowledge Discovery*, 11(1): 5–33, 2005.
- [2] Djellel Difallah, Elena Filatova, and Panos Ipeirotis. Demographics and dynamics of mechanical turk workers. In *Proceedings of the eleventh ACM international conference on web search and data mining*, pages 135–143, 2018.

- [3] Brian Everitt, Sabine Landau, Morven Leese, and Daniel Stahl. Cluster analysis. 2011.
- [4] Dominik Mautz, Wei Ye, Claudia Plant, and Christian Böhm. Towards an optimal subspace for k-means. In *Proceedings of the 23rd ACM SIGKDD International Conference on Knowledge Discovery and Data Mining*. ACM, aug 2017. doi: 10.1145/3097983.3097989. URL <https://doi.org/10.1145/3097983.3097989>.
- [5] Marieke E Timmerman, Eva Ceulemans, Kim De Roover, and Karla Van Leeuwen. Subspace k-means clustering. *Behav Res Methods*, 45(4):1011–23, Dec 2013. doi: 10.3758/s13428-013-0329-y.
- [6] Jérôme Waldispühl, Attila Szantner, Rob Knight, Sébastien Caisse, and Randy Pitchford. Leveling up citizen science. *Nat Biotechnol*, 38(10):1124–1126, 10 2020. doi: 10.1038/s41587-020-0694-x.
- [7] Rui Xu and Don Wunsch. *Clustering*, volume 10. John Wiley & Sons, 2008.

Appendix B

Human-supervised Clustering of Multidimensional Data using Crowdsourcing

Alexander Butyaev, Chrisostomos Drogaris, Olivier Tremblay-Savard & Jérôme Waldispühl

Answer to reviewers: Round #2

Again, we would like to thank the reviewers for the careful review of our manuscript and constructive comments. We do appreciate the time invested in reviewing our manuscript and positive feedback. We answer below the comments made in this second round of review.

Comments by Reviewer 3

You mentioned "100 participants who submitted over 1100 solutions" over a 2-week period, and I believe that 700 of those 1100 solutions were VRD related puzzles played by 75 participants.

Who were those participants in terms of ColonyB?

Was it anyone who happened to play during those 2 weeks?

Was it only those who reached a specific status on the leaderboard, or had played at least X number of games beforehand, or played ColonyB for a minimum number of days?

Participants to these experiments were regular players of ColonyB. We did not filter or select participants. This design intended to quantify the general performance of our mobile crowd (our app is only available on the Apple Store and Google Play), while simultaneously collecting as many data as possible in a short time-frame. Yet, we implemented filtration techniques to remove outliers such as the Average Cluster Size (ACS) technique described in the paper. This has been clarified in the revised manuscript.

When was this 2-week period and was it announced in-game that these puzzles would be any different than the regular ones, or on the contrary: did you intentionally not say anything?

We ran the experiment from March 3 to March 17, 2019. Intentionally, we did not make any specific announcement within or outside the game about this experiment, in order to not influence the behavior of the participants. Though, the later were aware from the beginning that the game could eventually include control experiments. This has been clarified in the revised manuscript.

How many participants have played ColonyB since it was posted on both app stores?

I ask because 100 participants sounds very low when reading the paper, but might be understandable based on my Question #1. It might help to mention that thousands of players have downloaded and tried ColonyB.

We agree this deserves a clarification. We never actively promoted the game after its initial launch (Aug. 25, 2016). Furthermore, our promotion campaign was limited to a press release from McGill University followed by an article in a local newspaper, and mentions on our social media channels. Occasionally, the website <https://scistarter.org/> highlighted our app in one of its blogs. As for most citizen projects (but also games), the participation follows a power law. Three years after the launch, the participation reached a steady state that is in-line with the statistics presented in this manuscript.

Since its launch in 2016, our application has been downloaded by 6,006 IOS users and 4,680 Android users. Thus, in total over 10,000 people tried ColonyB. Among them, 6433 of them completed the tutorial and 6123 submitted at least one solution. We are now including this information in our manuscript.

For completeness, we note that the annotations collected for the initial microbiome data set were not as conclusive as we hoped due to the high amount of noise and the absence of reference labels. We are still collecting annotations (at a slower pace) on improved data sets and hope that the result of this manuscript will provide us more confidence in subsequent analysis.

"We intentionally skipped AP and MeanShift due to their low average performance in the previous experiment."

I understand this, but did you actually run these two (and just decided not to plot the useless results) or did you never run them at all?

If it is quick to run them, it might be better to be able to state: "We ran AP and MeanShift, but omitted them from Figure X due to poor results, similar to their low average performance in the previous experiment,."

Indeed, we have run AP and MeanShift and omitted them due to lower than average performance in both cases. We added the suggested statement in the revised manuscript.

"During the development of Colony B, we observed that many players selected large clusters. We hypothesize that these players assumed that larger clusters receive a larger score, even though it is rather the opposite."

As you observed this during ColonyB's development, did you change the instructions in the tutorial to address this for future puzzles (such as EVE online)?

The instructions of ColonyB have not changed since its launch. We preferred integrating this observation in our analysis. As for Project discovery 3 (in Eve Online), we offer indeed a longer tutorial with clusters of various sizes which aims to address this issue. Furthermore, participants are not limited to one single cluster annotation and the score is not associated to the size of the clusters. All together, observations made in this study did help us to design a better tutorial for Project discovery 3.

Lastly, Figure 2 is very clear, but I have a suggestion: Panel D has a yellow circle (remaining from the previous screen) which is distracting... and is even faintly visible on Panel E.

Thanks for the suggestion. The yellow circles (with a number inside) are visual feedback given by the game to the players. These artifacts can remain for some time during the game until it fades away if the player tries to encircle data as quickly as possible. We edited the image to make it less distracting.

Comments by Reviewer 4

Comments to the Author(s)

The contributions of this paper are not apparent. The introduction of the paper is not sufficient to provide an apt summary of the problem to be solved and highlight the novelty of the work. A better explanation of the goal of the data analysis should be provided.

We regret that the objectives, methods and discussion were not as clear as we hoped. Since related comments were made in the previous round of reviews and have already been approved at this round, we prefer to not change drastically the abstract or introduction. We think the objective is already clearly defined in the introduction: *“The goal of this contribution is thus to offer a proof of concept that human-computing and crowdsourcing techniques can be applied to address the challenge of clustering abstract data with various sizes, shapes, and densities.”*. Yet, we tried to proceed to minor changes in the revised manuscript to address this comment.

In the introduction and related work, most literature are old. It is suggested to cover more recent works to show this area is active and absorbing. Some are suggested here: \Clustering structure analysis in time-series data with density-based clusterability measure," "Accelerated Two-Stage Particle Swarm Optimization for Clustering Not-Well-Separated Data," \Extracting Significant Mobile Phone Interaction Patterns based on Community Structures," "A Novel Rolling Bearing Vibration Impulsive Signals Detection Approach Based on Dictionary Learning."

We initially chose to cite seminal papers such as (Zhang, 1971) to highlight the fundamental nature of our work. Following reviewer advices from the previous round, we already added more recent references. Among the references suggested, we prefer to omit those focusing on time-series since our method is rather designed for detecting 2D Gestalt clusters.

Yet, we thank the reviewer for suggesting “Accelerated Two-Stage Particle Swarm Optimization for Clustering Not-Well-Separated Data“ which is indeed an appropriate citation. We also added others.

The technical content is not well described.

Following comments from reviewers #1 and #3 in the previous round, many improvements have already been made to the technical description of our methods and we believe it is now satisfying. We conjecture that the reviewer had only access to an earlier version of the manuscript, which we think could explain this comment (and probably others too).

The study has not explained well why one should consider the implemented methods over other alternatives, or what other alternatives are available. Some conventional machine learning methods have been used in this work for comparison. But they treat community detection as a problem of verifying the accuracy of clusters on large graphs. They generate a network consisting of a few dimensions to reconstruct the original network. However, this type of representation to a low dimensional space is linear. The fact that the real-world networks include nonlinear structures makes the traditional strategies less useful. As the scale of networks increases in the real world, more robust and efficient techniques are required to achieve high performance.

Our objective is primarily to demonstrate the relevance to capture the collective agreement from human crowd to improve clustering method. This is introduced in the second paragraph of the introduction and emphasized in its last sentence *“The goal of this contribution is thus to offer a proof of concept that human-computing and crowdsourcing techniques can be applied to address the challenge of clustering abstract data with various shapes and densities”*.

Indeed, this is not a method of choice yet (mostly due to the complexity of setting up the crowd-sourcing system) but this work offers a preliminary study for larger scale projects such as Project Discovery 3 in Eve Online mentioned in the revised version of the manuscript.

Unfortunately, we do not fully understand the comment about community detection “But they treat community detection as a problem of verifying the accuracy of clusters on large graphs”. The annotations from participants are assembled as a graph, and the Louvain algorithm is only used to identify community of players that may have consistent behaviors. It is only used as an input for our second method that showed lower performance. While we believe there is room for improvement, we acknowledge in this work that this approach did not obtained the best results.

Regarding the result section, the authors should notice that a separate dimensionality reduction phase is needed for using methods such as DBSCAN. It is obvious that without performing such a phase, the accuracy of traditional methods would degrade.

Our goal was to use DBSCAN and other off the shelf methods as a baseline to our work in order to interpret the results. Preprocessing the data for DBSCAN only could compromise the interpretability of our experiments and at the same time would require an extensive investigation/discussion of the settings and parameters used. For these reasons, we chose to not add this step.

If the authors aim to propose novel clustering methods, they should discuss the novelty of their algorithms and compare them with state-of-the-art methods over standard datasets.

As stated above, our goal is to demonstrate the relevance to capture the collective agreement from human crowd to improve clustering method. We aim to show the occurrence of a signal not exploited by other techniques. Our approach is intentionally simple and we do not aim compete with each single algorithm on the market. It would also require an in-depth comparison including discussion of settings and inclusion of many different data sets, which is beyond the scope of what our platform can currently do (see answers to comments from reviewer #1 and # 3 at the previous round). Finally, the section *Limitations* has been already extended at the previous round to discuss related issues.

It will be better if the computation time can be provided for the proposed algorithm and its competitors in experiments since the efficiency also matters. More statistics data (e.g. MSE) should be provided to support the point "GMM does not yield stable results (recall that we report the average score over 1000 runs)" as GMM performs satisfactorily in most cases.

A fair estimate of the empirical complexity of our methods should include the time require to collect the crowdsourcing data, which is irrelevant for fully-automated methods. At the previous round, we added a subsection *Scalability and Sustainability* that was perhaps not available to the reviewer at the time of the review and may already address this comment.

Appendix C

Human-supervised Clustering of Multidimensional Data using Crowdsourcing

Alexander Butyaev, Chrisostomos Drogaris, Olivier Tremblay-Savard & Jérôme Waldispühl

April 19, 2022

We thank the reviewers for the positive reviews of our manuscript and useful comments. As suggested, we have carefully checked our manuscript for grammatical errors and added the recommended citations in the introduction (Note: we added the references as early as possible in the manuscript to increase their visibility). Finally, we also deposited all the data and code in a public repository at <https://datadryad.org/stash/share/1RCFe5uhbSccq7DU0eMwXz1QwWyBf6Z6YMNWcgpXPMQ>.

Sincerely,

Jérôme Waldispühl
School of Computer Science
McGill University